# Composing Task-Agnostic Policies with Deep Reinforcement Learning

**Ahmed H. Qureshi**
UC San Diego
a1qureshi@ucsd.edu

**Jacob J. Johnson**
UC San Diego
jjj025@eng.ucsd.edu

**Yuzhe Qin**
UC San Diego
y1qin@eng.ucsd.edu

**Taylor Henderson**
UC San Diego
tjwest@ucsd.edu

**Byron Boots**
University of Washington
bboots@cs.washington.edu

**Michael C. Yip**
UC San Diego
yip@ucsd.edu

## Abstract

The composition of elementary behaviors to solve challenging transfer learning problems is one of the key elements in building intelligent machines. To date, there has been plenty of work on learning task-specific policies or skills but almost no focus on composing necessary, task-agnostic skills to find a solution to new problems. In this paper, we propose a novel deep reinforcement learning-based skill transfer and composition method that takes the agent's primitive policies to solve unseen tasks. We evaluate our method in difficult cases where training policy through standard reinforcement learning (RL) or even hierarchical RL is either not feasible or exhibits high sample complexity. We show that our method not only transfers skills to new problem settings but also solves the challenging environments requiring both task planning and motion control with high data efficiency.

## 1 Introduction

Compositionality is the integration of primitive functions into new complex functions that can further be composed into even more complex functions to solve novel problems (Kaelbling & Lozano-Pérez, 2017). Evidence from neuroscience and behavioral biology research shows that humans and animals have the innate ability to transfer their basic skills to new domains and compose them hierarchically into complex behaviors (Rizzolatti et al., 2001). In robotics, the primary focus is on acquiring new behaviors rather than composing and re-using the already acquired skills to solve novel, unseen tasks (Lake et al., 2017).

In this paper, we propose a novel policy ensemble composition method[1] that takes the basic, task-agnostic robot policies, transfers them to new complex problems, and efficiently learns a composite model through standard- or hierarchical-RL (Schulman et al., 2015; 2017; Haarnoja et al., 2018b; Vezhnevets et al., 2017; Florensa et al., 2017; Nachum et al., 2018). Our model has an encoder-decoder architecture. The encoder is a bidirectional recurrent neural network that embeds the given skill set into latent states. The decoder is a feed-forward neural network that takes the given task information and latent encodings of the skills to output the mixture weights for skill set composition. We show that our composition framework can combine the given skills both concurrently (*and*-operation) and sequentially (*or*-operation) as per the need of the given task. We evaluate our method in challenging scenarios including problems with sparse rewards and benchmark it against the state-of-the-art standard- and hierarchical- RL methods. Our results show that the proposed composition framework is able to solve extremely hard RL-problems where standard- and hierarchical-RL methods are sample inefficient and either fail or yield unsatisfactory results.

---

[1]Supplementary material and videos are available at https://sites.google.com/view/compositional-rl

## 2 RELATED WORK

In the past, robotics research has been primarily focused on acquiring new skills such as Dynamic Movement Primitives (DMPs) (Schaal et al., 2005) or standard reinforcement learning policies. A lot of research in DMPs revolves around learning compact, parameterized, and modular representations of robot skills (Schaal et al., 2005; Ijspeert et al., 2013; Paraschos et al., 2013; Matsubara et al., 2011). However, there have been quite a few approaches that address the challenge of composing DMPs in an efficient, scalable manner. To date, DMPs are usually combined through human-defined heuristics, imitation learning or planning (Konidaris et al., 2012; Muelling et al., 2010; Arie et al., 2012; Veeraraghavan & Veloso, 2008). Likewise, RL (Sutton & Barto, 2018) research is also centralized around learning new policies (**?**Schulman et al., 2015; 2017; Haarnoja et al., 2018b) for complex decision-making tasks by maximizing human-defined rewards or intrinsic motivations (Silver et al., 2016; Qureshi et al., 2017; 2018; Levine et al., 2016).

To the best of the authors' knowledge, there hardly exists approaches that simultaneously combine and transfer past skills into new skills for solving new complicated problems. For instance, Todorov (2009), Haarnoja et al. (2018a) and Sahni et al. (2017) require humans to decompose high-level tasks into intermediate objectives for which either Q-functions or policies are obtained via learning. The high-level task is then solved by merely maximizing the average intermediate Q-functions or combining intermediate policies through temporal-logic. Note that these approaches do not combine task-agnostic skills thus lack generalizability and the ability to transfer skills to the new domains. A recent and similar work to ours is a multiplicative composition policies (MCP) framework (Peng et al., 2019). MCP comprises i) a set of trainable Gaussian primitive policies that take the given state and proposes the corresponding set of action distributions and ii) a gating function that takes the extra goal information together with the state and outputs the mixture weights for composition. The primitive policies and a gating function trained concurrently using reinforcement learning. In their transfer learning tasks(Peng et al., 2019), the primitive polices parameters are kept fixed, and the gating function is trained to output the mixture weights according to the new goal information. In our ablation studies, we show that training an MCP like-gating function that directly outputs the mixture weights without conditioning on the latent encoding of primitive actions gives inferior performance compared to our method. Our method utilizes all information (states, goals, and primitive skills) in a structured way through attention framework, and therefore, leads to better performance.

Recent advancements lead to Hierarchical RL (HRL) that automatically decomposes the complex tasks into subtasks and sequentially solves them by optimizing the given objective function (**?**Vezhnevets et al., 2017; Nachum et al., 2018). In a similar vein, the options framework (Sutton et al., 1999; Precup, 2000) is proposed that solves the given task through temporal abstraction. Recent methods such as option-critic algorithm (Bacon et al., 2017) simultaneously learns a set of sub-level policies (options), their termination functions, and a high-level policy over options to solve the given problem. Despite being an exciting step, the option-critic algorithm is hard to train and requires regularization (Vezhnevets et al., 2016; Harb et al., 2018), or else it ends up discovering options for every time step or a single option for the entire task. In practice, the sub-level options or objectives obtained via HRL are inherently task-specific and therefore cannot be transferred to new domains.

## 3 BACKGROUND

We consider a standard RL formulation based on Markov Decision Process (MDP) defined by a tuple $\{\mathcal{S}, \mathcal{A}, \mathcal{P}, \mathcal{R}\}$, where $\mathcal{S}$ and $\mathcal{A}$ represent the state and action space, $\mathcal{P}$ is the set of transition probabilities, and $\mathcal{R}$ denotes the reward function. At time $t \geq 0$, the agent observes a state $s_t \in \mathcal{S}$ and performs an action $a_t \in \mathcal{A}$. The agent's action $a_t$ transitions the environment state from $s_t \in \mathcal{S}$ to $s_{t+1} \in \mathcal{S}$ with respect to the transition probability $\mathcal{P}(s_{t+1}|s_t, a_t)$ and leads to a reward $r_t \in \mathcal{R}$.

For compositionality, we extend the standard RL framework by assuming that the agent has access to the finite set of primitive policies $\Pi = \{\pi_i\}_{i=0}^N$ that could correspond to agent's skills, controller, or motor-primitives. Our composition model is agnostic to the structure of primitive policy functions, but for the sake of this work, we assume that each of the sub-policies $\{\pi_i\}_{i=0}^N$ solves the MDP defined by a tuple $\{\hat{\mathcal{S}}, \mathcal{A}, \hat{\mathcal{P}}, \hat{\mathcal{R}}^i\}$. Therefore, $\hat{\mathcal{S}}$, $\hat{\mathcal{P}}$ and $\hat{\mathcal{R}}^i$ are the state-space, transition probabilities and rewards of the primitive policy $\pi_i$, respectively. Each of the primitive policies $\pi_i : \hat{S} \times \mathcal{A} \rightarrow [0, 1], \forall i \in [0, 1, \cdots, N]$, takes a state $\hat{s} \in \hat{\mathcal{S}}$ and outputs a distribu-

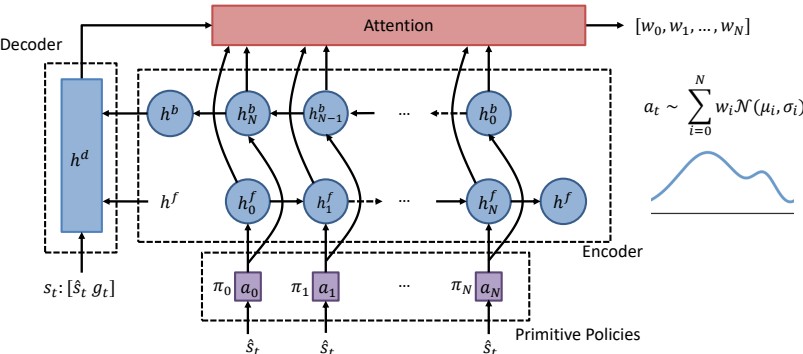

Figure 1: Policy ensemble composition model that takes the state information $s_t$ and a set of primitive policies' output $\{\hat{a}_i\}_{i=0}^N$ to compute a composite action $a_t$.

tion over the agent's action space $\mathcal{A}$. We define our composition model as a composite policy $\pi_\theta^c : \mathcal{S} \times \mathcal{A}^{N+1} \times \mathcal{A} \to [0, 1]$, parameterize by $\theta$, that outputs a distribution over the action space conditioned on the environment's current state $s \in \mathcal{S}$ and the primitive policies $\{\hat{a}_i \in \mathcal{A}\}_{i=0}^N \sim \Pi$. The state space of the composite model is $\mathcal{S} = [\hat{\mathcal{S}}, \mathcal{G}]$. The space $\mathcal{G}$ could include any task specific information such as target locations. Hence, in our framework, the state inputs to the primitive policies $\Pi$ and composite policy $\pi_\theta^c$ need not to be the same.

In remainder of this section, we show that our composition model solves an MDP problem. To avoid clutter, we assume that both primitive policy ensemble and composite policy have the same state space $\mathcal{S}$, i.e., $\mathcal{G} = \emptyset$. The composition model samples an action from a distribution parameterized by the actions of sub-level policies and the state $s \in \mathcal{S}$ of the environment. Therefore, we can augment the given state space $\mathcal{S}$ as $\mathcal{S}^c : \mathcal{A}^{N+1} \times \mathcal{S}$, where $\mathcal{A}^{N+1} : \{\mathcal{A}^i\}_{i=0}^N$ are the outputs of sub-level policies. Hence, compositional MDP is defined as $\{\mathcal{S}^c, \mathcal{A}, \mathcal{P}^c, \mathcal{R}\}$ where $\mathcal{S}^c = \mathcal{A}^N \times \mathcal{S}$ is the new composite state-space, $\mathcal{A}$ is the action-space, $\mathcal{P}^c : \mathcal{S}^c \times \mathcal{S}^c \times \mathcal{A} \to [0, 1]$ is the transition probability function, and $\mathcal{R}$ is the reward function for the given task.

## 4 POLICY ENSEMBLE COMPOSITION

In this section, we present our policy ensemble composition framework, shown in Fig. 1. Our composition model consists of i) the encoder network that takes the outputs of primitive policies and embeds them into latent spaces; ii) the decoder network that takes current state $s_t$ of the environment and the latent embeddings from the encoder network to parameterize the attention network; iii) the attention network that outputs the probability distribution over the primitive low-level policies representing their mixture weights. The remainder of the section explains the individual models of our composition framework and the overall training procedure.

### 4.1 ENCODER NETWORK

Our encoder is a bidirectional recurrent neural network (BRNN) that consists of Long Short-Term Memory units (Hochreiter & Schmidhuber, 1997). The encoder takes the outputs of the policy ensemble $\{\hat{a}_i\}_{i=0}^N$ and transform them into latent states of forward and backward RNN, denoted as $\{h_i^f\}_{i=0}^{N+1}$ and $\{h_i^b\}_{i=0}^{N+1}$, respectively, where $h_i^f, h_i^b \in \mathbb{R}^d; \forall i \in [0, 1, \cdots, N + 1]$. The $N + 1$ states of forward and backward RNN corresponds to their last hidden states denoted as $h^f$ and $h^b$, respectively, in Fig. 1.

### 4.2 DECODER NETWORK

Our decoder is a simple feed-forward neural network that takes the last hidden states of the forward and backward encoder network, i.e., $\{h^f, h^b\}$, and the current state of the environment $s$ to map them into a latent space $h \in \mathbb{R}^d$. The state input to the decoder network is defined as $s : [\hat{s}, g]$,

where $\hat{s} \in \hat{\mathcal{S}}$ is the state input to the low-level policy ensemble and $g \in \mathcal{G}$ could be any additional information related to the given task, e.g., goal position of the target to be reached by the agent.

## 4.3 ATTENTION NETWORK

The composition weights (see Fig. 1) $\{w_i \in [0,1]\}_{i=0}^N$ are determined by the attention network as follows:

$$q_i = W^T \cdot \tanh(W_f \cdot h_i^f + W_b \cdot h_i^b + W_d \cdot h); \forall i \in [0, N] \tag{1}$$

where $W_f, W_b, W_d \in \mathbb{R}^{d \times d}$ and $W \in \mathbb{R}^d$. The weights $\{w_i\}_{i=0}^N$ for the composite policy are computed using gumbel-softmax denoted as $\mathrm{softmax}(q/T)$, where T is the temperature term (Jang et al., 2016).

## 4.4 COMPOSITE POLICY

Given the primitive policy ensemble $\Pi = \{\pi_i\}_{i=0}^N$, the composite action is the weighted sum of all primitive policies outputs, i.e., $\pi_\theta^c = \sum_i^N w_i \pi_i$. Since, we consider the primitive policies to be Gaussian distributions, the output of each primitive policy is parameterized by mean $\mu$ and variance $\sigma$, i.e., $\{\hat{a}_i \sim \mathcal{N}(\mu_i, \sigma_i)\}_{i=0}^N \leftarrow \{\pi_i\}_{i=0}^N$. Hence, the composite policy can be represented as $\pi_\theta^c = \sum_i^N w_i \mathcal{N}(\mu_i, \sigma_i)$, where $\mathcal{N}(\cdot)$ denotes Gaussian distribution, and $\sum_i w_i = 1$. Given the mixture weights, other types of primitive policies, such as DMPs (Schaal et al., 2005), can also be composed together by the weighted combination of their normalized outputs.

## 4.5 COMPOSITE MODEL TRAINING OBJECTIVE

The general objective of RL methods is to maximize the cumulative expected reward, i.e., $J(\pi_\theta^c) = \mathbb{E}_{\pi_\theta^c}[\sum_{t=0}^\infty \gamma^t r_t]$, where $\gamma : (0,1]$ is a discount factor. We consider the policy gradient methods to update the parameters $\theta$ of our composite model, i.e., $\theta \leftarrow \theta + \eta \bigtriangledown_\theta J(\pi_\theta^c)$, where $\eta$ is the learning rate. We show that our composite policy can be trained through standard RL and HRL methods, described as follow.

### 4.5.1 STANDARD REINFORCEMENT LEARNING

In standard RL, the policy gradients are determined by either on-policy or off-policy updates (?Schulman et al., 2015; 2017; Haarnoja et al., 2018b) and any of them could be used to train our composite model. However, in this paper, we consider off-policy soft-actor critic (SAC) method (Haarnoja et al., 2018b) for the training of our policy function. SAC maximizes the expected entropy $\mathcal{H}(\cdot)$ in addition to the expected reward, i.e.,

$$J(\pi_\theta^c) = \sum_{t=0}^T \mathbb{E}_{\pi_\theta^c}[r(s_t, a_t) + \lambda \mathcal{H}(\pi_\theta^c(\cdot|s_t))] \tag{2}$$

where $\lambda$ is a hyperparameter. We use SAC as it motivates exploration and has been shown to capture the underlying multiple modes of an optimal behavior. Since there is no direct method to estimate a low-variance gradient of Eq (2), we use off-policy value function-based optimization algorithm (for details refer to Appendix A.1 of supplementary material).

### 4.5.2 HIERARCHICAL REINFORCEMENT LEARNING

In HRL, there are currently two streams - task decomposition through sub-goals (Nachum et al., 2018) and option framework (Bacon et al., 2017) that learns temporal abstractions. In the options framework, the options can be composite policies that are acquired with their termination functions. In task decomposition methods that generate sub-goal through high-level policy, the low-level policy can be replaced with our composite policy. In our work, we use the latter approach (Nachum et al., 2018), known as HIRO algorithm, to train our policy function.

Like, standard HIRO, we use two level policy structure. At each time step $t$, the high-level policy $\pi_{\theta'}^{hi}$, with parameters $\theta'$, observes a state $s_t$ and takes an action by generating a goal $g_t \in \mathcal{S}$ in the state-space $\mathcal{S}$ for the composite low-level policy $\pi_\theta^{c:low}$ to achieve. The $\pi_\theta^{c:low}$ takes the state

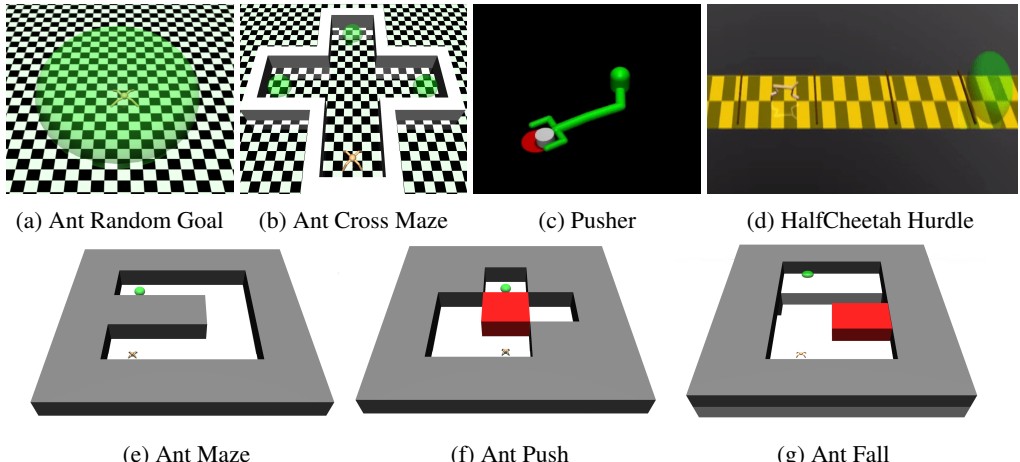

Figure 2: Benchmark control and manipulation tasks requiring an agent to reach or move the object to the given targets (shown in red for pusher and green for rest).

$s_t$, the goal $g_t$, and the primitive actions $\{\hat{a}_i\}_0^N$ to predict a composite action $a_t$ through which an agent interacts with the environment. The high-level policy is trained to maximize the expected task rewards given by the environment whereas the composite low-level policy is trained to maximize the expected intrinsic reward defined as the negative of distance between current and goal states, i.e., $\|s_t + g_t - s_{t+1}\|^2$. To conform with HIRO settings, we perform off-policy correction of the high-level policy experiences and we train both high- and low-level policies via TD3 algorithm (Fujimoto et al., 2018) (for details refer to Appendix A.2 of supplementary material).

## 5 EXPERIMENTS AND RESULTS

We evaluate and compare our method against standard RL, and HRL approaches in challenging environments (shown in Fig. 2) that requires complex task planning and motion control. The implementation details of all presented methods and environment settings are provided in Appendix C of supplementary material. We also do an ablative study in which we take away different components of our composite model to highlight their importance. Furthermore, we depict attention weights of our model in a navigation task to highlight its ability of concurrent and sequential composition.

We consider the following seven environments for our analysis: (1) **Pusher:** A simple manipulator has to push an object to a given target location. (2) **Ant Random Goal:** In this environment, a quadruped-Ant is trained to reach the randomly sampled goal location in the confined circular region. (3) **Ant Cross Maze:** The cross-maze contains three target locations. The task for a quadruped Ant is to reach any of the three given targets by navigating through a 3D maze without collision. (4) **HalfCheetah Hurdle:** In this problem, the task for a halfcheetah is to run and jump over the three barriers to reach the given target location. (5) **Ant Maze:** A ⊃-shaped maze poses a challenging navigation task for a quadruped-Ant. In this task, the agent is given random targets all along the maze to reach while training. However, during the evaluation, we test the agent for reaching the farthest end of the maze. (6) **Ant Push:** A challenging environment that requires both task and motion planning. The environment contains a movable block, and the goal region is located behind that block. The task for an agent is to reach the target by first moving to the left of the maze so that it can move up and right to push the block out of the way for reaching the target. (7) **Ant Fall:** A navigation task where the target is located across the rift in front of the agent's initial position. There also happen to be a moveable block, so the agent has to move to the right, push the block forward, fill the gap, walk across, and move to the left to reach the target location. (8) **Multi-goal Point Mass:** In this scenario, the task is to navigate a point-mass to one of the four goals located diagonally to agent initial position.

In all tasks, we also acquire primitive skills of the agent for our composite policy. For Ant, we use four basic policies for moving left, right, up, and down. The pusher uses two primitive policies that are to push an object to the left and down. In HalfCheetah hurdle environment, the low-level policies include jumping and running forward. Finally fot the point-mass robot, the composition model takes

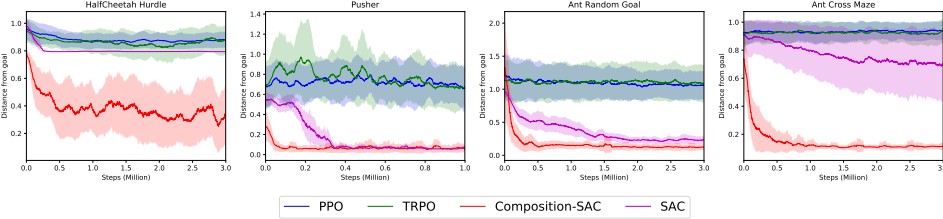

Figure 3: Comparison results of our method against several standard RL methods averaged over ten trials in a set of difficult tasks. The vertical and horizontal axis represents the distance of the agent/object from the target and environment steps in millions, respectively. Note that our composition framework learns to solve the task with high samples efficiency, whereas other benchmark methods either fail or perform poorly.

four policies for moving in the up, down, left and right directions. Furthermore, in all environments, except pusher, the primitive policies were agnostic of high-level tasks ( such as target locations) that were therefore provided separately to our composite model via decoder network. This highlights the ability of our model to transfer basic robot skills to novel problems.

| Methods | Environments | | | |
|---|---|---|---|---|
| | Ant Random Goal | Ant Cross Maze | Pusher | HalfCheetah Hurdle |
| SAC | $0.21 \pm 0.08$ | $0.78 \pm 0.06$ | $0.17 \pm 0.02$ | $0.79 \pm 0.01$ |
| TRPO | $1.09 \pm 0.15$ | $0.85 \pm 0.15$ | $0.64 \pm 0.09$ | $0.87 \pm 0.05$ |
| PPO | $1.06 \pm 0.11$ | $0.95 \pm 0.07$ | $0.71 \pm 0.06$ | $0.88 \pm 0.04$ |
| **Our Method** | $\mathbf{0.11 \pm 0.05}$ | $\mathbf{0.11 \pm 0.02}$ | $\mathbf{0.14 \pm 0.02}$ | $\mathbf{0.27 \pm 0.22}$ |

Table 1: Performance comparison of our model against SAC (Haarnoja et al., 2018b), TRPO (Schulman et al., 2015), and PPO (Schulman et al., 2017) on benchmark control tasks in terms of distance (lower the better) of an agent from the given target. The mean final distances with standard deviations over ten trials are reported. We also normalize the reported values by the agent initial distance from the goal so values close to 1 or higher show failure. It can be seen that our method (shown in bold) accomplishes the tasks by reaching goals whereas other methods fail except for SAC in simple Pusher and Ant Random Goal environments.

## 5.1 COMPARATIVE STUDY

In our comparative studies, we divide our test environments into two groups. The first group includes Pusher, Random Goal Ant, Ant Cross Maze, and HalfCheetah-Hurdle environments, whereas the second group comprises the remaining environments that require task and motion planning under weak reward signals.

In the first group of settings, we compare our composite model trained with SAC (Haarnoja et al., 2018b) against the standard Gaussian policies obtained using SAC (Haarnoja et al., 2018b), PPO (Schulman et al., 2017), and TRPO (Schulman et al., 2015). We exclude HRL methods in these cases as the environment rewards sufficiently represent the underlying task, whereas HRL approaches are applicable in cases that have a weak reward signal or require task and motion planning. Table 1 presents the mean and standard deviation of the agent's final distance from the given targets after the end of an evaluation rollout over the ten trials. Fig. 3 shows the mean learning performance over all trials during the three million training steps. In these set of problems, TRPO and PPO entirely fail to reach the goal, and SAC performs reasonably well but only in simple Ant Random Goal and Pusher environments as it fails in other cases. Our composite policy obtained using SAC successfully solves all tasks and exhibit high data-efficiency by learning in merely a few thousand training steps.

In our second group of environments, we use distance-based rewards that are weak signals as greedily following them does not lead to solving the problem. Furthermore, in these environments, policies trained with standard RL, including our composite policy, failed to solve the problem even after 20 million training steps. Therefore, we trained our composite policy with HIRO (Nachum et al., 2018) and compared its performance against standard HIRO formulation (Nachum et al., 2018). We also tried to include option-critic framework (Bacon et al., 2017), but we were unable to get any

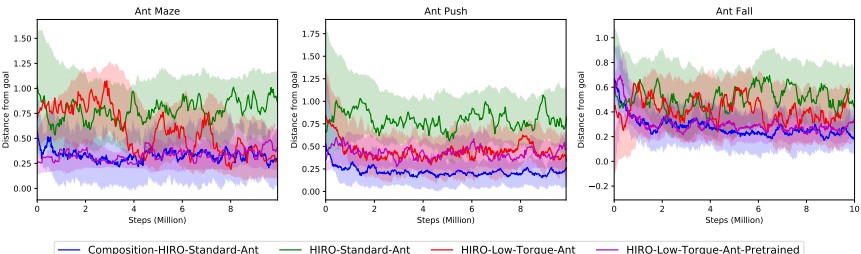

Figure 4: Performance comparison of our composition model trained with HIRO on a standard Ant (150 units torque limit) against three variants of standard HIRO formulation in three challenging environments. The pretrained HIRO model undergoes 4 million steps of extra training to counter for the training time utilised by premitive skills of our composition model. The low-torque-Ant has 30 units toruqe limit. We report mean and standard error, over ten trials, of agent's final distances from the given goals, normalized by their initial distance, over 10 million steps.

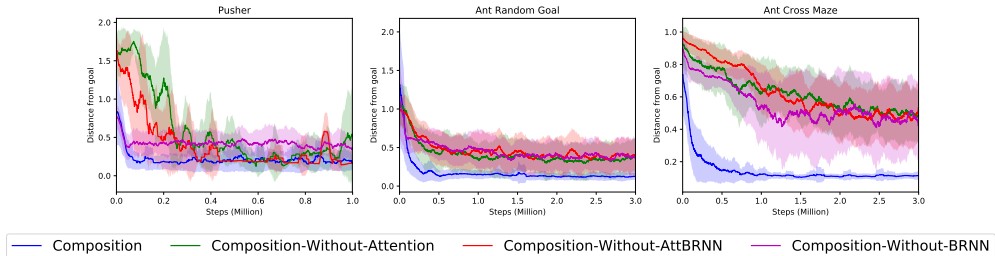

Figure 5: Ablative Study: Performance comparison, averaged over ten trials, of our composite model against its ablated variations that lack attention model, bidirectional-RNN (BRNN) or both attention and BRNN (AttBRNN) in three different environments.

considerable results with their online implementation despite several attempts with the parameter tuning. One of the reasons option-critic fails is because it relies purely on task rewards to learn, which makes them inapplicable for cases with weak reward signals (Nachum et al., 2018).

Nachum et al. (2018) use a modified Ant in their paper that has 30 units joint torque limit (low-torque-Ant). For our composition method, we use Mujoco standard Ant that has a torque limit of 150 units which makes the learning even harder as the Ant is now more prone to instability than a low-torque-Ant. For standard HIRO formulation, we trained three variants i) HIRO trained on standard Mujoco Ant, ii) HIRO trained on low-torque Ant (as in (Nachum et al., 2018)); iii) HIRO pretrained for the equal number of training steps as used to train the primitive policies of our composition method on a low-torque-Ant (For more details, refer to Appendix C.1).

Fig. 4 shows the learning performance, averaged over ten trials, during 10 million steps. It can be seen that the composite policy with HIRO outperforms standard HIRO (Nachum et al., 2018) by a significant margin. It also certifies the utility of solving RL tasks using composition by leveraging basic pre-acquired skills. Furthermore, HIRO performs poorly with standard Ant, even pretraining did not improve the performance, as it imposes a harder control problem.

## 5.2 ABLATIVE STUDY

We remove bidirectional RNN (BRNN) (similar to (Peng et al., 2019)), attention-network, and both attention-network and BRNN (AttBRNN) from our composition model to highlight their importance in the proposed architecture in solving complex problems. We train all models with SAC (Haarnoja et al., 2018b). The first model is our composite policy without attention in which the decoder network takes the state information and last hidden states of the encoder (BRNN) to directly output actions rather than mixture weights. The second model is without attention network and BRNN, it is a feed-forward neural network that takes the state information and the primitive actions and predicts the action to interact with the environment. The third model is without BRNN, it is a feed-forward neural network that takes the state and goal information, and output the mixture weights. The mixture weights are then used to combine the actions from primitive policies. This setting is

same as (Peng et al., 2019) for their transfer learning problems. Fig. 5 shows the mean performance comparison, over ten trials, of our composite model against its ablated versions on a Ant Random Goal, Cross Maze Ant, and Pusher environment. We exclude remaining test environments in this study as ablated models completely failed to perform or show any progress.

Note that the better performance of our method compared to ablated versions highlight the merits of our architecture design. Intuitively, BRNN allows the dynamic encoding of a skill set that could be of variable lengths[2]. And our decoder network uses the encoded skills together with given state and goal information (states, goals, and primitive skills) in a structured way using attention framework and provides significantly better performance than other ablated models, including (Peng et al., 2019).Furthermore, another merit of using the attention network is that it bypasses the complex transformation of action embeddings (composition-without-attention) or actions and state-information (composition-without-AttBRNN) directly to action space. Hence, the proposed architecture is crucial for the composition of task-agnostic sub-level policies to solve new problems.

### 5.3 DEPICTION OF ATTENTION WEIGHTS

In order to further assess the merit of utilizing an attention network, we apply our model to a simple 2D multi-goal point-mass environment as shown in Fig. 6. The point-mass is initialized around the origin (with variance $\sigma^2 = 0.1$) and must randomly choose one of four goals to reach. For this experiment we use dense rewards with both a positional and actuation cost. Primitive policies of *up* $(+y)$, *down* $(-y)$, *left* $(-x)$, and *right* $(+x)$ were trained and composed to reach goals, represented here as red dots, in the "diagonal" directions where a combination of two or more primitive policies are required to reach each goal.

The four mappings in the figure give us insight into how the attention network is utilizing the given primitives to achieve the desired task. At each step in a given path, the weights $\{w_i\}_{i=0}^N$ for each low-level policy are assigned and composed together to move the point-mass in the

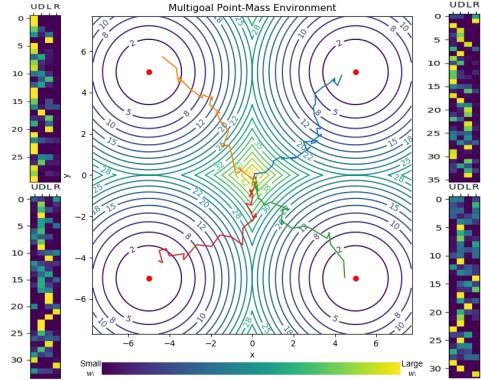

Figure 6: Each path corresponds to its adjacent attention weight mapping. The weighting "strength" of each primitive policy is depicted for each step (i.e. up (U), down (D), left (L), and right (R)). Each path begins at the origin and ends when the point-mass is within one unit of a goal. The plot contours represent the position cost.

desired direction. We see here that even with some noise and short-term error, the attention weights are strongest for primitive policies that move the point-mass to its chosen goal. We also see that multiple policies are activated at once to achieve more direct movements toward the goal, as opposed to "stair-stepping" where only one primitive is activated at a time. Both of these observations point to the concurrent and sequential nature of this composition model. Please refer to Appendix D for depiction of attention weights in other complicated environments.

## 6 CONCLUSIONS AND FUTURE WORK

We present a novel policy ensemble composition method that combines a set of independent and task-agnostic primitive policies through reinforcement learning to solve the given tasks. We show that our method can transfer the given skills to novel problems and can compose them both sequentially (*or* -operation) and concurrently (*and* -operation) to find a solution for the task in hand. Our experiments highlight that composition is vital for solving problems requiring complex motion skills and decision-making where standard reinforcement learning and hierarchical reinforcement learning methods either fail or need a massive number of interactive experiences to achieve the desired results.

In our future work, we plan to extend our method to automatically acquire the missing skills in the given skillset that are necessary to solve the specific problems (refer to Appendix B for prelimi-

---

[2]However, in this paper we only consider a primitive skill set of fixed size

nary results). We also aim towards a system that learns the hierarchies of composition models by combining primitive policies into complex policies that would further be composed together for a combinatorial outburst in the agent's skillset.

ACKNOWLEDGMENTS

We would like to thank Ben Eysenbach for sharing his halfcheetah-hurdle environment implementation.

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

---

**Algorithm 1:** Composition model training using SAC

---

Initialize parameter vectors $\phi, \phi', \theta, \xi$
**Input:** Primitive policies $\Pi = \{\pi_i\}_{i=0}^N$
**for** *each iteration* **do**
    **for** *each environment step* **do**
        Compute primitive policies state $\hat{s}_t \leftarrow s_t \backslash g_t$
        Sample primitive actions $\{\hat{a}_{i,t}\}_{i=0}^N \sim \Pi(\hat{s}_t)$
        Sample composite action $a_t \sim \pi_\theta^c(a_t|s_t, \{\hat{a}_{i,t}\}_{i=0}^N)$
        Sample next state $s_{t+1} \sim p(s_{t+1}|s_t, a_t)$
        $\mathcal{M} \leftarrow \mathcal{M} \cup \{(s_t, a_t, \{\hat{a}_{i,t}\}_{i=0}^N, r_t, s_{t+1})\}$
    **for** *each gradient step* **do**
        Update value function $\phi \leftarrow \phi - \eta \bigtriangledown_\phi J_V(\phi)$
        Update Q-function $\xi \leftarrow \xi - \eta \bigtriangledown_\xi J_Q(\xi)$
        Update policy $\theta \leftarrow \theta - \eta \bigtriangledown_\theta J_{\pi^c}(\theta)$
        $\phi' \leftarrow \tau\phi + (1-\tau)\phi'$

---

## A   Composite Model Training Algorithms

### A.1   Training with Soft Actor-Critic

In this section, we briefly describe the procedure to train our composition model using SAC (Haarnoja et al., 2018b). Although any RL method can be used to optimize our model, we use SAC as it is reported to perform better than other training methods. Our composite policy is a tractable function $\pi_\theta^c(a_t|s_t, \{\pi_i\}_{i=0}^N)$ parameterized by $\theta$. The composite policy update through SAC requires the approximation of Q- and value-functions. The parametrized value- and Q-function are denoted as $V_\phi(s_t)$ with parameters $\phi$, and $Q_\xi(s_t, a_t)$ with parameters $\xi$, respectively. Since, SAC algorithm build on the soft-policy iteration, the soft value-function $V_\phi(s_t)$ and soft Q-function $Q_\xi(s_t, a_t)$ are learned by minimizing the squared residual error $J_V(\phi)$ and squared Bellman error $J_Q(\xi)$, respectively, i.e.,

$$J_V(\phi) = \mathbb{E}_{s_t \sim \mathcal{M}}[\frac{1}{2}(V_\phi(s_t) - \hat{V}(s_t))^2] \tag{3}$$

$$J_Q(\xi) = \mathbb{E}_{(s_t, a_t) \sim \mathcal{M}}[\frac{1}{2}(Q_\xi(s_t, a_t) - \hat{Q}(s_t, a_t))^2] \tag{4}$$

where $\mathcal{M}$ is a replay buffer, $\hat{V}(s_t) = \mathbb{E}_{a_t \sim \pi_\theta^c}[Q_\xi(s_t, a_t) - \log \pi_\theta^c(a_t|s_t)]$ and $\hat{Q}$ is the Bellman target computed as follows:

$$\hat{Q}(s_t, a_t) = r(s_t, a_t) + \gamma \mathbb{E}_{s_{t+1} \sim p}[V_{\phi'}(s_t + 1)] \tag{5}$$

The function $V_{\phi'}(s_t)$ is the target value function with parameters $\phi'$. The parameters $\phi'$ are the moving average of the parameters $\phi$ computed as $\tau\phi + (1-\tau)\phi'$, where $\tau$ is the smoothing coefficient. Finally the policy parameters are updated by minimizing the following expected KL-divergence.

$$J_{\pi^c}(\theta) = \mathbb{E}_{s_t \sim \mathcal{M}}\left[D_{KL}\left(\pi_\theta^c(\cdot|s_t)\big|\big|\frac{\exp(Q_\xi(s_t, \cdot))}{Z_\xi(s_t)}\right)\right] \tag{6}$$

where $Z_\xi$ is a partition function that normalizes the distribution. Since, just-like SAC, our Q-function is differentiable, the above cost function can be determined through a simple reparametization trick, see Haarnoja et al. (2018b) for details. Like SAC, we also maintain two Q-functions that are trained independently, and we use the minimum of two Q-functions to compute Eqn. 3 and Eqn. 6. This way of using two Q-function has been shown to alleviate the positive biasness problem in the policy improvement step. The overall training procedure is summarized in Algorithm 1.

### A.2   Training with HIRO

In this section, we outline the algorithm to train composite policy through HIRO that employs the two level policy structure. The high-level policy generates the sub-goals for the low-level composite

---

**Algorithm 2:** Composition model training using HIRO

---

Initialize parameter vectors $\phi$, $\phi'$, $\theta$, $\xi$
**Input:** Primitive policies $\Pi = \{\pi_i\}_{i=0}^N$
**for** *each iteration* **do**
    **for** *each environment step* **do**
        Compute primitive policies state $\hat{s}_t \leftarrow s_t \backslash g_t$
        Sample primitive actions $\{\hat{a}_{i,t}\}_{i=0}^N \sim \Pi(\hat{s}_t)$
        Sample high-level action $g_t \sim \pi^{hi}(\hat{s}_t)$
        Sample composite action $a_t \sim \pi_\theta^c(a_t | s_t, g_t, \{\hat{a}_{i,t}\}_{i=0}^N)$
        Sample next state $s_{t+1} \sim p(s_{t+1} | s_t, a_t)$
        $\mathcal{M} \leftarrow \mathcal{M} \cup \{(s_t, g_t, a_t, \{\hat{a}_{i,t}\}_{i=0}^N, r_t, s_{t+1})\}$
    **for** *each gradient step* **do**
        Sample mini-batch with c-step transitions
        $\{(g^k, s_{t:t+c}^k, a_{t:t+c-1}^k, \{\hat{a}_{j,t:t+c-1}^k\}_{j=0}^N, r_{t:t+c-1}^k)\}_{k=1}^B \sim \mathcal{M}$
        Compute rewards for low-level policy $\{r_i^{lo}\}_{i=0}^B \leftarrow \{r^{lo}(s_i, g_i, s_{i+1})\}_{i=0}^B$
        Update $\pi^{c:lo}$ w.r.t $Q^{lo}$ using $\{(s_k, g_k, a_k, \{\hat{a}_{i,k}\}_{i=0}^N, r_k^{lo}, s_{k+1})\}_{k=0}^{B-1}$ (Nachum et al., 2018)
        Update $\pi^{hi}$ w.r.t $Q^{hi}$ using $\{(s_k, \hat{g}_k, \sum_{i=k}^{c-1} r_k, s_{k+c})\}_{k=0}^{B-1}$ (Nachum et al., 2018)

---

policy to achieve. The low-level composite policy also have access to the primitive policy actions. Like HIRO, we use TD3 algorithm (Fujimoto et al., 2018) to train both high-level and low-level policies with their corresponding Q-functions, $Q^{hi}$ and $Q^{lo}$, respectively. The low-level policy $\pi_\theta^{c:low}$, with parameters $\theta$, is trained to maximize the Q-values from the low-level Q-function $Q^{lo}$ for the given state-goal pairs. The Q-function ($Q^{lo}$) parameters $\xi$ are optimized by minimizing temporal-difference error for the given transitions, i.e.,

$$J_Q^{lo}(\xi) = \left( r^{lo}(s_t, g_t, s_{t+1}) + \gamma Q_\xi^{lo}(s_{t+1}, g_{t+1}, \pi_\theta^{c:lo}(s_{t+1}, g_{t+1}, \{\hat{a}\}_0^N)) - Q_\xi^{lo}(s_t, g_t, a_t) \right)^2 \quad (7)$$

where $r^{lo}(s_t, g_t, s_{t+1}) = -\|s_t + g_t - s_{t+1}\|$ and $g_{t+1} \sim \pi_{\theta'}^{hi}(s_{t+1})$.

The high-level policy $\pi_{\theta'}^{hi}$, with parameters $\theta'$, is trained to maximize the values of $Q^{hi}$. The Q-function ($Q^{hi}$) parameters $\xi'$ are trained through minimizing the following loss for the given transitions.

$$J_Q^{hi}(\xi') = \left( \sum_{t=0}^{c-1} R_t(s_t, a_t, s_{t+1}) + \gamma Q_{\xi'}^{hi}(s_{t+c}, \pi_{\theta'}^{hi}(s_{t+c})) - Q_\xi^{hi}(s_t, \hat{g}_t) \right)^2 \quad (8)$$

During training, the continuous adaptation of low-level policy poses a non-stationery problem for the high-level policy. To mitigate the changing behavior of low-level policy, Nachum et al. (2018) introduced off-policy correction of the high-level actions. During correction, the high-level policy action $g$ is usually re-labeled with $\hat{g}$ that would induce the same low-level policy behavior as was previously induced by the original high-level action $g$ (for details, refer to (Nachum et al., 2018)). Algorithm 2 presents the procedure to train our composite policy with HIRO.

# B  FUTURE RESEARCH DIRECTIONS & DISCUSSION

In this section, we present preliminary results to highlight the potential research avenues to further extend our method. Apart from compositionality, another research problem in the way of building intelligent machines is the autonomous acquisition of new skills that were lacking in the system and therefore, hindered it from the solving the given task (Lake et al., 2017). In this section, we demonstrate that our composition model holds the potential for building such a system that can simultaneously learn new missing skills and compose them together with the existing skill set to solve the given problem.

Fig. 7 shows a simple ant-maze environment in which the 3D quadruped-Ant has to reach the target, indicated as a green region. In this problem, we equip our composition model with a single primitive

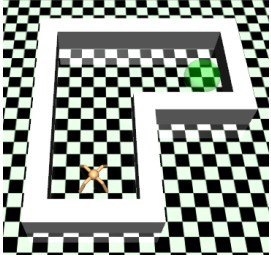 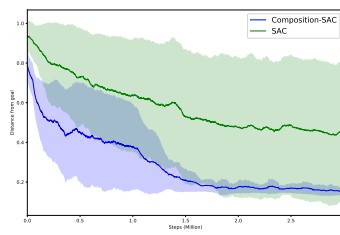

Figure 7: New skill acquisition task. The composite model has access to a primitive policy for moving right and a trainable policy function. Our method trained the new function to move in the upward direction to reach the given goal.

policy for walking in the right direction and a trainable, parametric policy function. The trainable policy function takes the state without goal information and outputs the parameters for Gaussian distribution to model the robot's action space. Note that the composite model requires a skill of walking upward in addition to moving in the right direction to reach the given goal. To determine if our composition model can teach the new parametric policy function to move upward, we trained our composition model along with the new policy using the shared Q- and value-functions. Once trained, we observed that our composition model learned the parametric policy function for walking in an upward direction with slight right-turning behavior.

The plot in Fig. 7 shows the performance of our composition model and standard RL policy obtained with SAC in this environment. The vertical axis indicates the distance from the target location averaged over five randomly seeded trials. The horizontal axis shows the number of environment steps in millions. It can be observed that our method converged faster than standard RL policy. It shows that our approach is also utilizing the existing skills and therefore learns faster than a standard policy function that solves the entire task from scratch. Furthermore, in current settings, we do not impose any constraint on the new policy function that would make it learn only the missing skills rather than replicating existing skills or solving the given task entirely by itself. We leave the formulation of such constraint for autonomous acquisition of new, missing skills to our future works.

## C   IMPLEMENTATION DETAILS

### C.1   PRETRAINING BENCHMARK METHOD

In our results, we also consider the pretraining of benchmark methods for an equivalent amount of time as used to train the primitive policies for our composition model for a better analysis. The total number of steps needed to train primitive skills of Ant (up, down, left, and right), halfcheetah (run and jump), and pusher (down and left) were 1, 0.5, and 0.1 million, respectively. In Fig. 4, HIRO was pretrained for both standard (30 units) and low-torque-ant (150 units). The results of HIRO standard-ant are excluded in Fig.4 to avoid clutter as they led to no improvement in the performance. For the non-hierarchical tasks presented in Fig. 3, the benchmark methods are not pretrained. However, to account for pretraining, the performance of other methods (TRPO, PPO, SAC) can be accessed after 1, 0.5, and 0.1 million for the ant, halfcheetah, and pusher environments, respectively. Also, notice that in Fig. 3, the pretraining will have no significant effect on the performance of TRPO and PPO in all environments and SAC in half-cheetah hurdle and cross-maze-ant environments. Furthermore, since pusher and random-goal-ant are relatively simple environments (due to no obstacles or maze around), pretrained SAC can perform similar to our composition method.

### C.2   ENVIRONMENT DETAILS

In this section, we present the environment details including reward functions, primitive policies, and state space information. The reward functions are presented in the Table 3 together with the overall reward scaling values. For Ant and halfcheetah-hurdle environments, the primitive policies were trained in a free-space that had no mazes, hurdle or fixed targets in the surrounding.

### C.2.1 ANT ENVIRONMENTS

In these environments, we use 8 DOF four-legged Ant with 150 units torque limit. The primitive policies of moving left, right, down and up were shared across all these tasks. These sub-level-policies were obtained using SAC and each policy was trained for about 0.25 million steps. In these environments, the information $g$ in the state $s : [\hat{s}, g]$ corresponds to the target location. Let us introduce the notation to defined reward function. Let $r_{xy}$, $g_{xy}$, $u$, and $f_c$ denote xy-position of the robot's torso, xy-position of the goal, joint torques, and contact-cost, respectively. The scaling factors are defined as $\lambda$. The reward function for the following environments is defined as with reward scaling of 5 units:

$$-\lambda_g||r_{xy} - g_{xy}||^2 + \lambda_v v_{xy} + \lambda_s I(\text{IsAlive}) - \lambda_{ct}||u||^2 - \lambda_c f_c \qquad (9)$$

**Ant Random Goal:** In this environment, the ant has to navigate to any randomly sampled target within the confined circular region of radius 5 units. The goal radius is defined to be 0.25 units. The reward function coefficients $\lambda_g$, $\lambda_v$, $\lambda_s$, $\lambda_{ct}$, and $\lambda_c$ are 0.3, 0.0, 0.05, 0.01, and 0.001, respectively.

**Ant Cross Maze:** In this environment, the ant has to navigate through the 3D maze to reach any of the target sampled from the three targets. The goal radius is defined to be 1.0 units. The reward function parameters are same as for the random-goal ant environment.

For the remaining environment (Ant Maze, Ant Push and Ant Fall), we use the following reward function with no reward scaling:

$$\lambda_g||r_{xyz} - g_{xyz}||^2 - \lambda_{ct}||u||^2 - \lambda_c f_c \qquad (10)$$

where coefficients $\lambda_g$, $\lambda_{ct}$, and $\lambda_c$ are set to be 1.0, 0.05, and $0.5 \times 10^{-4}$.

**Ant Maze:** In this environment, we place the Ant in a $\supset$-shaped maze for a navigation task between given start and goal configurations. The goal radius is defined to be 5 units. During training, the goal is uniformly sampled from $[-4, 20] \times [-4, 20]$ space, and the Ant initial location is always fixed at $(0, 0)$. During testing, the agent is evaluated to reach the farthest end of the maze located at $(0, 19)$ within L2 distance of 5.

**Ant Push:** In this environment, the Ant is initially located at $(0, 0)$ coordinate, the moveable block is at $(0, 8)$, and the goal is at $(0, 19)$. The agent is trained to reach randomly sampled targets whereas during testing, we evaluate the agent to reach the goal at $(0, 19)$ within L2 distance of 5.

**Ant Fall:** In this environment, the Ant has to navigate in a 3D maze. The initial agent location is $(0, 0)$, and a movable block is at $(8, 8)$ at the same elevation as Ant. Their is a rift in the region $[-4, 12] \times [12, 20]$. To reach the target on the other side of the rift, the Ant must push the block down into the rift, and then step on it to get to the goal position.

| Parameters | SAC | HIRO | TRPO | PPO |
|---|---|---|---|---|
| Learning rate ($\eta$) | $3 \times 10^{-4}$ | $1 \times 10^{-4}$ | - | - |
| Discount factor ($\gamma$) | 0.99 | 0.99 | 0.99 | 0.99 |
| Nonlinearity in feedforward networks | ReLU | ReLU | ReLU | ReLU |
| Minibatch samples size | 256 | 128 | - | - |
| Replay buffer size | $10^6$ | $2 \times 10^5$ | - | - |
| Batch-size | - | - | 1000 | 1000 |
| Target parameters smoothing coefficient ($\tau$) | 0.005 | 0.005 | - | - |
| Target parameters update interval | 1 | 2 | - | - |
| Gradient steps | 1 | 1 | 0.01 | 0.01 |
| Gumbel-softmax temperature ($T$) | 0.5 | 0.5 | - | - |

Table 2: Hyperparameters

### C.2.2 PUSHER

In pusher environment, a simple manipulator has to move an object to the target location. The primitive policies were to push the object to the bottom and left. These low-level policies were obtained via SAC after 0.1 million training steps. In this environment, the state information for both primitive policies and the composite policy include the goal location. Therefore, $\mathcal{G}$, in this case, is null. The reward function is given as:

$$-\lambda_g||o_{xy} - g_{xy}||^2 - \lambda_o||r_{xy} - o_{xy}||^2 - \lambda_{ct}||u||^2 \qquad (11)$$

| Model Architectures | | Hidden units |
|---|---|---|
| Composition-HIRO | High-level Policy: Three layer feed forward network | 300 |
| | Encoder Network: Bidirectional RNN with LSTMs | 128 |
| | Decoder Network (Single layer feed forward network) | 128 |
| | Attention Network: $W_f, W_b, W_d \in \mathbb{R}^{d \times d}; W \in \mathbb{R}^d$ | 128 |
| Composition-SAC | Encoder Network: Bidirectional RNN with LSTMs | 128 |
| | Decoder Network (Single layer feed forward network) | 128 |
| | Attention Network: $W_f, W_b, W_d \in \mathbb{R}^{d \times d}; W \in \mathbb{R}^d$ | 128 |
| HIRO | High-level Policy: Three layer feed forward network | 300 |
| | Low-level Policy: Three layer feed forward network | 300 |
| Standard RL policy | Two layer feed forward network | 256 |

Table 3: Network Architectures.The right most column shows the hidden units per layer.

where $o_{xy}$, $g_{xy}$, $r_{xy}$, and $u$ are xy-position of object, xy-position of goal, xy-position of arm, and joint-torques. The coefficients $\lambda_g$, $\lambda_o$, and $\lambda_{ct}$ are 1.0, 0.1, and 0.1, respectively.

### C.2.3 HALFCHEETAH-HURDLE

In halfcheetah-hurdle environment, a 2D cheetah has to jump over the three hurdles to reach the target. The primitive policies include run forward and jump where each poilcy was trained with SAC for 0.25 million steps. Furthermore, in this environment, the information $g$ in the state $s : [\hat{s}, g]$ corresponds to the x-position of the next nearest hurdle in front of the agent as well as the distance from that hurdle. The reward function is defined as:

$$-\lambda_g||r_{xy} - g_{xy}||^2 - \lambda_{hc}hc(\cdot) + \lambda_{rg}I(\text{goal}) + \lambda_z|v_z| + \lambda_v v_x - \lambda_c cc(\cdot) \tag{12}$$

where $r_{xy}$, $g_{xy}$, $v_z$, and $v_x$ are xy-position of robot torso, xy-position of goal, velocity along z-axis, and velocity along x-axis, respectively. The function $hc(\cdot)$ returns a count indicating the number of hurdles in front of the robot. The indicator function $I(\text{goal})$ returns 1 if the agent has reached the target otherwise 0. The function $cc(\cdot)$ is a collision checker which returns 1 if the agent collides with the hurdle otherwise 0. The reward function coefficients $\lambda_g$, $\lambda_{hc}$, $\lambda_{rg}$, $\lambda_z$, $\lambda_v$, and $\lambda_c$ are 0.1, 1.0, 1000, 0.3, 1.0 and 2, respectively.

### C.3 HYPERPARAMETERS AND NETWORK ARCHITECTURES

Table 2 summarizes the hyperparameters used to train policies with SAC (Haarnoja et al., 2018b), TRPO (Schulman et al., 2015), PPO (Schulman et al., 2017), and HIRO (Nachum et al., 2018). Table 3 summarizes the network architectures.

## D DEPICTION OF ATTENTION WEIGHTS

In this section, we illustrate the attention weights of our composite policy in halfcheetah-hurdle (Fig. 9), pusher (Fig. 10), and cross-maze-ant (Fig. 11) environments. Our results highlight the ability of our model to compose a given set of primitive polices, both sequentially and concurrently, to solve the given problem. Fig. 8 shows the color-coding for the scale of attention weight value given by the composite policy.

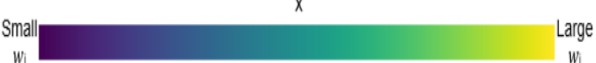

Figure 8: The attention weights: The blue and yellow colors show low and high values, respectively

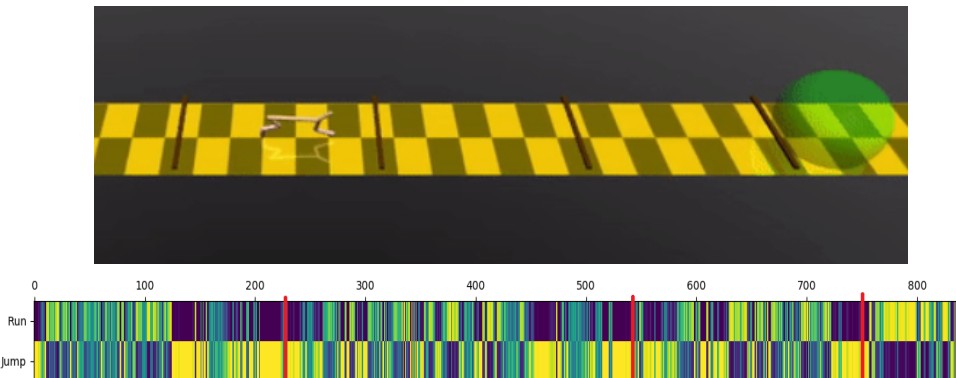

Figure 9: The depiction of attention weights for the halfcheetah-hurdle environment. The weighting "strength" of each primitive policy (i.e., run and jump) for each step is shown. There are three hurdles in front of the agent. The red-markers on the attention plot shows the position of those hurdles. It can be seen that the agent concurrently mixes the jump and run primitives before approaching the next hurdle to gain momentum and switches to jump primitive when passing over the hurdles.

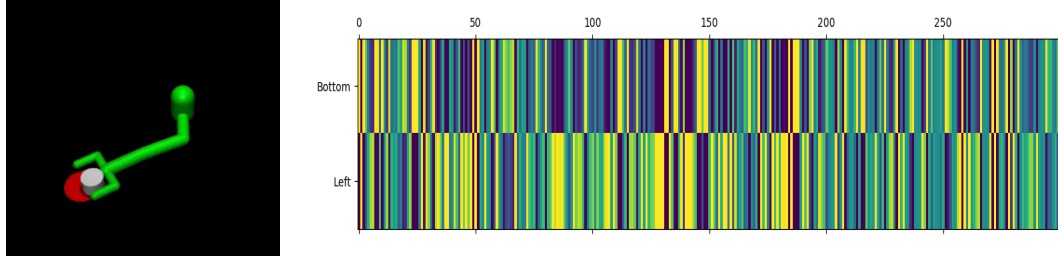

Figure 10: The depiction of attention weights for the pusher environment. The weighting "strength" of each primitive policy (i.e., Bottom and Left) for each step is shown. It can be seen that the agent concurrently as well sequentially compose the push-to-the-bottom and push-to-the-left primitives to reach the target.

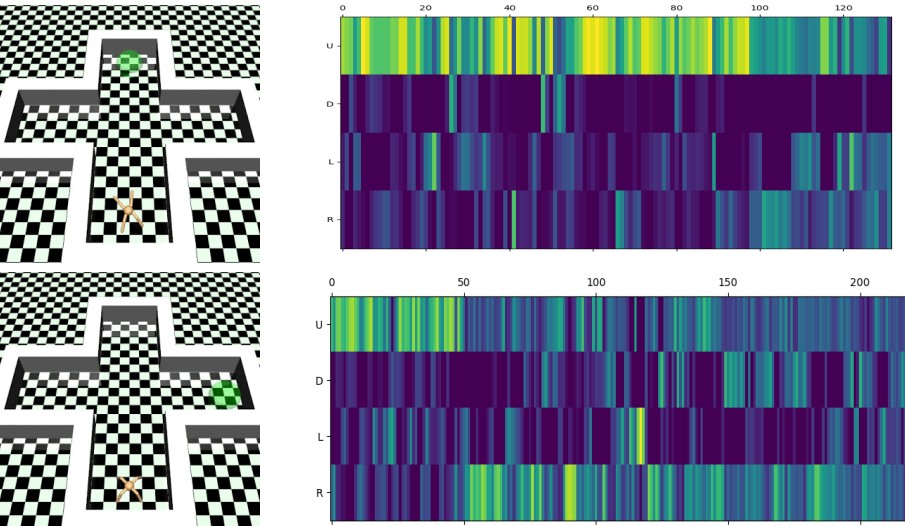

Figure 11: The depiction of attention weights for the cross-maze environment. The weighting "strength" of each primitive policy (i.e., up (U), down (D), left (L), and right (R)) for each step is shown. The top and bottom rows show the agent reaching the goals on the top and right, respectively. In the top row, the agent emphasizes on the upward primitive and learns to ignore the remaining primitives. In the bottom row, the agent first focuses on the upward primitive to reach the cross-point and then switches to right-direction motion primitive to reach the given goal.

