# OpenReview forum: "Composing Task-Agnostic Policies with Deep Reinforcement Learning"
_ICLR.cc/2020/Conference — Accept (Poster)_

### Official Review · AnonReviewer2 · 2019-10-19
**Official Blind Review #2**

**Rating:** 6

**Review:**

Overall, I think the method has some great merit but I am not overly confident in the reproducibility of the method. Some of the comparisons (HIRO) do not agree with the results in the HIRO paper. Also, more description is needed to describe how the baselines were used in the analysis. Were they also given the pre-trained sub-policies? A more fair comparison might be to give those baseline methods no sub-policies but let them pre-train for an equivalent amount of time as the sub-policies are trained.

 Here are some more detailed comments:
- Figure 1 is not very clear and does not appear to add much to the explanation of the method. More detail should be included in the caption.
 - In the paper, it is noted that HRL has high sample complexity and needs lots of training data. I find the comment about how HRL requires many more training steps than regular RL very odd. The purpose of HRL is to have better sample efficiency and learn strong polices faster. Has the author observed different? The purpose of HRL is to reduce sample complexity and search in a well suited and structured way.
- The assumption that the sub-policies solve the underlying MDP is rather strong. How are these policies going to be trained to guarantee this?
- I like the idea of using an attention model to help pick learn a weighting for the combination of a number of sub-policies. I am not sure if using a bi-directional LSTM is the best or simplest method to accomplish this. The authors can look at "MCP: Learning Composable Hierarchical Control with Multiplicative Compositional Policies" NeurIPS 2019 for a recent work that is similar to theirs.
- For Figure 4: Do these methods also get to use the sub-policies that have pre-trained some version of tasks? Also, how are these component policies trained? Over what tasks are they trained? This information is very important to make sure the method is not overly biased to the composition of them.
- I find the results in Figure 5 very odd. The baseline shows that HIRO does not learn to perform well on these tasks even though these are the tasks from the HIRO paper that it learned to solve rather well. Can this contradiction be explained?
- For the HIRO comparison was the system also using the composite policies there were pretrained? HIRO is designed to learn the sub-policies concurrently but it seems in this case the authors are using the outputs of the composition policies as input to the HIRO low policy.
- I do not understand the visualization in Figure 7. How to the colored paths for the agent represent the weights for the compose policy?

**Experience Assessment:**

I have published in this field for several years.

**Review Assessment: Checking Correctness Of Derivations And Theory:**

I carefully checked the derivations and theory.

**Review Assessment: Checking Correctness Of Experiments:**

I carefully checked the experiments.

**Review Assessment: Thoroughness In Paper Reading:**

I read the paper thoroughly.

---

> ### Author Response · Authors · 2019-11-11
> **Response to Reviewer # 2**
>
> RC: reviewer comment; AR: author response
>
> We provide the source code to ensure reproducibility with trained polices.  Also, we were able to reproduce HIRO results presented in their paper. In their paper, they use low-torque-ant (30 Units) that limits the action space. In contrast, in our work, we use standard-ant (150 units), so the action-space is large and makes the learning problem significantly harder as the Ant is now more prone to instability. For completeness, we now update for plots (Fig. 4) to include HIRO results with low-toque-ant (conforming with the system settings proposed in their paper).
>
> RC: Also, more description is needed to describe how the baselines were used in the analysis. Were they also given the pre-trained sub-policies? A more fair comparison might be to give those baseline methods no sub-policies but let them pre-train for an equivalent amount of time as the sub-policies are trained.
>
> AR: We have included more descriptions on the baselines’ analysis in Appendix C. Furthermore, we now include HIRO baselines that were pretrained on low-torque-ant (as in HIRO paper) for the equal amount of time as the sub-policies for the composite model were trained. The new results are presented in Fig 4. The pretraining of HIRO on standard-Ant does not lead to any improvement. Therefore, to avoid clutter, we do not include those plots in Fig 4.  Furthermore, we would like to highlight that, in the case of Ant, our composition model uses the same primitive skills across all presented environments, which in some cases is not equivalent to pretraining other benchmark models every time.
>
> RC: Figure 1 is not very clear and does not appear to add much to the explanation of the method.
>
> AR: We exclude figure 1 as other reviewers think it’s unnecessary.
>
> RC: In the paper, it is noted that HRL has high sample complexity and needs lots of training data. I find the comment about how HRL requires many more training steps than regular RL very odd. The purpose of HRL is to have better sample efficiency and learn strong polices faster. Has the author observed different? The purpose of HRL is to reduce sample complexity and search in a well suited and structured way.
>
> AR: We agree with the reviewer. Our statement was misleading. HRL methods perform indeed better than regular RL, and we have removed that statement. As per the author’s understanding, the HRL, especially HIRO, is for complex tasks with weak reward signals that require both task and motion planning. And, due to weak reward signals and complex decision-making, these methods take a huge amount of interactive experience with the environment, which, of course, is still much less than regular RL methods. And in this aspect, our proposed composition framework further improves data-efficiency of HIRO by effectively exploiting the primitive skills.
>
> RC: The assumption that the sub-policies solve the underlying MDP is rather strong. How are these policies going to be trained to guarantee this?
>
> AR: We assume standard RL sub-level-policies policies that generally solve underlying MDPs.
>
>
> RC: The authors can look at “MCP: Learning Composable Hierarchical Control with Multiplicative Compositional Policies” NeurIPS 2019 for a recent work that is similar to theirs.
>
> AR: Thanks for pointing us to MCP framework, we have included it in our related work section.
>
>
> RC: For Figure 3: Do these methods also get to use the sub-policies that have pre-trained some version of tasks? Also, how are these component policies trained? Over what tasks are they trained?
>
> AR: We have included the details of training sub-level policies in Appendix C.
>
> RC: I find the results in Figure 4 very odd. The baseline shows that HIRO does not learn to perform well on these tasks even though these are the tasks from the HIRO paper that it learned to solve rather well. Can this contradiction be explained?
>
> AR: We were able to reproduce HIRO results using their system settings (low-torque-ant), please refer to Fig 4 for results.
>
> RC: For the HIRO comparison was the system also using the composite policies there were pretrained? HIRO is designed to learn the sub-policies concurrently but it seems in this case the authors are using the outputs of the composition policies as input to the HIRO low policy.
>
> AR: In our proposed work, HIRO concurrently learns the high-level policy and a composite policy (given primitive skills).
>
> RC: I do not understand the visualization in Figure 6?
>
> AR: The “color” of paths has nothing to do with the attention weights. The agent starts from the center (origin) and moves towards one of the given goals. Since primitive polices were to move straight up, down, left, and right, the diagonal motion requires the concurrent composition of low-level skills, which is indicated by the attention weights. On attention maps, the vertical axis reads from top to down, where zero means the starting position of the agent, and it takes about 30 steps to reach the target.

---

> > ### Comment · AnonReviewer2 · 2019-11-13
> > **Thank you for your clarifications**
> >
> > Thank you for your clarifications
> >
> > These comments have helped clear up my understanding of some important details.
> >
> > It would be good to include more details on why your method is novel and an improvement over MCP. I think it is but the included comment does not highlight the importance of the distinction "our method does not focus on learning reusable policies; instead, it learns to compose standard task-agnostic RL policies"
> >
> > "Figure 3: Do these methods also get to use the sub-policies that have pre-trained some version of tasks? "
> > I did not find this information in the appendix. Could you summarize this process?

---

> > > ### Comment · AnonReviewer1 · 2019-11-13
> > > **MCP discussion**
> > >
> > > R1 here (not authors).
> > > I share the concern about MCP. However, while MCP has been accepted to NeurIPS, it has not yet been "published", and so it should maybe be considered "concurrent work" rather than prior work when evaluating novelty.

---

> > > > ### Comment · AnonReviewer2 · 2019-11-14
> > > > **MCP is concurrent work**
> > > >
> > > > Hello,
> > > >
> > > > I agree completely that MCP is "concurrent work". I just think it is helpful for the community if some nice info is included in the paper to discuss the similarities and differences.

---

> > > ### Author Response · Authors · 2019-11-14
> > > **Response to Reviewer # 2**
> > >
> > > We thank our reviewer for providing comprehensive and constructive feedback which helped us improve our paper. We have revised the paper to address reviewer comments. The response summaries are as follow:
> > >
> > > RC: It would be good to include more details on why your method is novel and an improvement over MCP.
> > >
> > > AR: We want to thank our reviewer for pushing us to go through the details of the MCP [1] paper carefully. We noticed that their approach is quite similar (if not the same) to our Composition-Without-BRNN model presented in the ablation study section (Section 5.2). We already show in our results that MCP like framework gives inferior performance compared to our method.  Further details are included in the paper (in related work section 2 and ablative study section 5.2) and are also summarized as follows:
> > >
> > > A recent and similar work to ours is a multiplicative composition policies (MCP) framework [1]. MCP comprises i) a set of trainable Gaussian primitive policies that take the given state and proposes the corresponding set of action distributions and ii) a gating function that takes the extra goal information together with the state and outputs the mixture weights for composition. The primitive policies and a gating function trained concurrently using reinforcement learning. In their transfer learning tasks [1], the primitive polices parameters are kept fixed, and the gating function is trained to output the mixture weights according to the new goal information. In our ablation studies, we show that training an MCP like-gating function that directly outputs the mixture weights without conditioning on the latent encoding of primitive actions gives inferior performance compared to our method. Our method utilizes all information (states, goals, and primitive skills) in a structured way through attention framework, and therefore, leads to better performance.
> > >
> > > [1] “MCP: Learning Composable Hierarchical Control with Multiplicative Compositional Policies” NeurIPS 2019
> > >
> > >
> > > RC:  "Figure 3: Do these methods also get to use the sub-policies that have pre-trained some version of tasks? " I did not find this information in the appendix. Could you summarize this process?
> > > AR: We include a separate section in Appendix (C.1) to discuss pretraining. A brief summary for Fig. 3 is as follow:
> > > For the non-hierarchical tasks presented in Fig. 3, the benchmark methods are not pretrained. However, to account for pretraining, the performance of other methods (TRPO, PPO, SAC) can be accessed after 1, 0.5, and 0.1 million for the ant, halfcheetah, and pusher environments, respectively. Also, notice that in Fig. 3, the pretraining will have no significant effect on the performance of TRPO and PPO in all environments and SAC in half-cheetah hurdle and cross-maze-ant environments. Furthermore, since pusher and random-goal-ant are relatively simple environments (due to no obstacles or maze around), pretrained SAC can perform similar to our composition method.

---

> > > > ### Comment · AnonReviewer2 · 2019-11-14
> > > > **Comparison appreciated**
> > > >
> > > > Thank you for the extra discussion. This helps highlight the papers novelty, improvement and connection to related work.

---

### Official Review · AnonReviewer1 · 2019-10-23
**Official Blind Review #1**

**Rating:** 6

**Review:**

What is the specific question/problem tackled by the paper?
This paper addresses the hierarchical RL problem of combining multiple primitive policies (pi_1, …, pi_K) into policies for more complex tasks. Given a number of primitive skills and a new task within an environment, the paper aims to learn to pick and combine the primitives as needed to solve the new task. This problem statement is interesting and the method performs well on difficult tasks.

However, I argue for rejecting this paper because it lacks meaningful contributions to the field. I do not see how the method presented in the paper is more than RL over  hand engineered action spaces that are better for the tasks. While this improves results, we already know that for any task, there is some best action space for performing that task. This is why most HRL work aims to also find the primitive policies in additional to composing them. Is this any better than option-critic if the options are hardcoded to be the primitives? The experiment state that the option-critic method did not work, but did you give it access to the same primitives?

Summary:
The method presented in the paper is as follows: at each state s_t, the K primitives are queried for their action a_k ~ pi_k(s_t). Then, a biRNN reads in the actions in order from 1 to k. In parallel, the state and a goal are encoded by a network named “Decoder”. The encoded state and the hidden states of the RNN are used to output an attention weight over each primitive. Finally, the output action is the weighted combination of all the actions. The encoders and attention weights are trained with RL.

This method is evaluated on several mujoco tasks, such as making a cheetah jump hurdles by combining “forward” and “jump” primitives. Each environment has predefined primitives such as “forward” “left” “right” etc.

This method is compared against HIRO, which does not have access to the primitive policies. It is not surprising that hand engineering primitives helps performance.

Is the approach well motivated?
The general idea behind the approach is well motivated: using primitive skills to learn complex skills is a useful goal. The details of the method are strange.
I would like to see a better motivation and empirical justification for the biRNN. Why should the primitive’s action be encode in order? The ordering of the primitives is arbitrary and constant: a fully connected network could be used, or the attentions could be output entirely independently per primitive.
In fact, I do see not why the primitives’ actions need to be encoded at all. It would be much simpler for the encoder to look at (s_t, g_t) and output a discrete probability over the K primitives.
The ablations in 5.2 are for outputting actions directly rather than mixture weights. The paper would benefit from ablations where mixture weights are output but without the biRNN or without passing in the primitive’s actions.


Is the method well placed in the literature?
The main idea of predicting weights over multiple experts is not novel (see "Adaptive mixtures of local experts” from 1991). In the context of RL literature, we can interpret the primitive skills as actions directly, and then the method is performing basic RL over a better action space (the better actions being “go left”, “jump” etc. We can also interpret these as options, but unlike options a single primitive is not followed for multiple time steps with a termination condition. Functionally, this is equivalent to regular RL using domain knowledge to engineer the action space.


**Experience Assessment:**

I have published in this field for several years.

**Review Assessment: Checking Correctness Of Derivations And Theory:**

N/A

**Review Assessment: Checking Correctness Of Experiments:**

I assessed the sensibility of the experiments.

**Review Assessment: Thoroughness In Paper Reading:**

I read the paper thoroughly.

---

> ### Author Response · Authors · 2019-11-11
> **Response to Reviewer # 1**
>
> RC: Reviewer Comment; AR: Authors Response
>
> RC: Is this any better than option-critic if the options are hardcoded to be the primitives? The experiment state that the option-critic method did not work, but did you give it access to the same primitives?
>
> AR: Option-critic method executes the options sequentially, and there is no concurrent composition. Furthermore, option-critic learns/needs tasks specific primitive, whereas our method can compose task-agnostic skills, both sequentially and concurrently, to solve new problems.
>
> RC: Why should the primitive’s action be encode in order?
>
> AR: We use bidirectional RNN instead of uni-directional RNN to avoid ordering issues. It is also evident from the results that ordering is not a problem for our method. For instance, in cross-maze-ant, the composition has access to four primitive policies ( up, down, left, and right), and still, our method learns to not use downward policy (irrespective of order) at all as it is not needed to solve the given task.
>
> RC: The ablations in 5.2 are for outputting actions directly rather than mixture weights. The paper would benefit from ablations where mixture weights are output but without the biRNN or without passing in the primitive’s actions.
>
> AR: We have included the suggested ablation. Please refer to Fig 5. Composition-without-BRNN takes the current state and outputs the mixture weights directly, which are then used to compose primitive actions. The composition-without-BRNN indeed perform better than other ablated models but still gives inferior performance than our proposed composition model. We argue that the composition model without BRNN performs poorly compared to the proposed method because it is entirely unaware of low-level continuous actions. Therefore, learning the latent encoding of primitive actions and making them a part of state-space is crucial for learning an effective composite policy.

---

> > ### Comment · AnonReviewer1 · 2019-11-13
> > **Thank you for your reply and for the updates to the paper.**
> >
> > Thank you for your reply and for the updates to the paper. The new result in Appendix B is interesting. For the new ablations, I would like some clarifications:
> > “Intuitively, BRNN allows the dynamic composition of a skill set of variable lengths. Therefore, the composition model without BRNN performs poorly compared to the proposed method as it is unaware of low-level continuous actions.”
> > None of the experiments use skill sets of variable lengths, right? These experiments each has between 2 and 4 primitive policies. It’s also quite strange that all the ablations perform the same as each other. Why would removing the attention lead to exactly the same result as removing the BRNN? Is there some local minimum at 0.5 distance to goal that they all get stuck in? I’m concerned about these numbers.
> >
> > “averaged over ten trials” Is a trial an evaluation rollout, or is a trial an independent RL training process?
> >
> > AR: We use bidirectional RNN instead of uni-directional RNN to avoid ordering issues. It is also evident from the results that ordering is not a problem for our method. For instance, in cross-maze-ant, the composition has access to four primitive policies ( up, down, left, and right), and still, our method learns to not use downward policy (irrespective of order) at all as it is not needed to solve the given task.
> > -> BiRNN still have ordering… I don’t find this decision to be well motivated, but it is what it is. Given that the BiRNN is there, yet could be interesting to see experiments where different primitives are available at different places/time within an environment, and so leverage the variable input length of the BiRNN.
> >
> >
> > Perhaps my view of the paper is best summarized by the paper itself: “MCP (Peng 2019) learns reusable primitive skills that can be combined and transferred to solve new problems. However, our method does not focus on learning reusable policies; instead, it learns to compose standard task-agnostic RL policies for solving new tasks.” To me, this does not sound like an ICLR paper. And, given the existence of MCP, which can either learn primitives or use pertained primitives, what is the motivation for this paper?
> >
> > A version of the paper that I would have been more excited about would have
> > 1. Motivated the use of hand-defined primitives, and/or performed experiments in domains where these already exist.
> > 2. Don’t almost all comparisons be against methods that do not have any access to the primitives (and are therefore more general but also slower to learn), these comparisons are obviously unfair.
> >     1. The new result that HIRO performs as well as Composition HIRO when given the same number of training steps shows this well, unless the low-torque also significantly improves the Composition HIRO (HIRO-Low-Torque-Ant-Pretrained).
> > 3. Spent a lot more of the paper trying out the obvious ways of using these primitives and discussing the design choices here. How did you end up with the architecture you have?

---

> > > ### Author Response · Authors · 2019-11-14
> > > **Response to Reviewer# 1**
> > >
> > > We would like to thank our reviewer for the feedback. We have revised our paper accordingly and our responses are summarized as follows:
> > >
> > > RC: None of the experiments use skill sets of variable lengths, right?
> > > AR: Yes, in this paper we use a fixed set of primitive policies and we further elaborate this in our paper to avoid confusion.
> > >
> > > RC: Is there some local minimum at 0.5 distance to goal that they all get stuck in?
> > > AR: Due to environment structures, the models are likely to get trapped in some local minima. For instance, in the ant-cross-maze, getting to the center junction is almost half of the distance (~0.5/1.0). Also, reaching any of the three goals requires upward motion to the center junction followed by further sideways or upward movements. We observed that all ablated models overfit to upward motion policy and fail to learn other motions. Similarly, in the ant-random-goal environment, we noticed that all ablated models could only acquire limited behaviors that helped them get closer to the goals limited to one-quarter or two-quarters of a circle but not the entire space.  We believe the presented environments are challenging and using advance exploration strategies might help improve the performance for all models, but it is not in the scope of this paper.
> > >
> > > RC: It’s also quite strange that all the ablations perform the same as each other?
> > > AR: Ablated models indeed fail to perform and sometimes converge quickly to some local minima, as we have mentioned earlier such as for the ant-cross-maze. However, it is also evident that learning curves are not the same for all models. For instance, in ant-cross-maze, the variance of each plot indicates that composition-without-BRNN was able to get closer to the goal sometimes, whereas, other ablated models completely failed to get any closer to the given targets.
> > >
> > > RC: “averaged over ten trials” Is a trial an evaluation rollout, or is a trial an independent RL training process?
> > > AR: They correspond to independent RL training processes.
> > >
> > > RC: And, given the existence of MCP, which can either learn primitives or use pertained primitives, what is the motivation for this paper?
> > > AR:  We further elaborate on MCP in our related works (section 2) to highlight the merit of our approach. And our response is also summarized as follows:
> > >
> > > MCP [1] comprises i) a set of trainable Gaussian primitive policies that take the given state and proposes the corresponding set of action distributions and ii) a gating function that takes the extra goal information together with the state and outputs the mixture weights for composition. The primitive policies and a gating function are trained concurrently using reinforcement learning. In their transfer learning tasks [1], the primitive polices parameters are kept fixed, and the gating function is trained to output the mixture weights according to the new goal information. In our ablation studies, we show that training an MCP like-gating function that directly outputs the mixture weights without conditioning on the latent encoding of primitive actions gives inferior performance compared to our method. Our method utilizes all information (states, goals, and primitive skills) in a structured way through attention framework, and therefore, leads to better performance.
> > >
> > > [1] “MCP: Learning Composable Hierarchical Control with Multiplicative Compositional Policies” NeurIPS 2019
> > >
> > > RC: Don’t almost all comparisons be against methods that do not have any access to the primitives, these comparisons are obviously unfair.
> > >
> > > AR: To address this issue, we include a comparison with the benchmark models that were pretrained for an equal amount of time that we used to train our primitive policies (Fig. 4 Appendix C.1). Also, note that in our framework, primitive policies are trained only once, especially for Ant, and are used to solve in all related environments (ant-cross-maze, ant-random-goal, ant-u-maze, ant-push, ant-fall).
> > >
> > > RC: HIRO performs as well as Composition HIRO, unless the low-torque also significantly improves the Composition HIRO (HIRO-Low-Torque-Ant-Pretrained).
> > >
> > > AR: It does not matter much for the composition framework either it is low-torque-ant or high-torque-ant as it assumes the availability of primitive skills. However, for HIRO standard-ant, we found that even pretraining led to no improvement in the performance whereas our method can use standard ant to solve the problem.
> > >
> > > AC: How did you end up with the architecture you have?
> > > We believe our ablation studies sufficiently motivates the architecture design. In our paper, we highlight that it is crucial to utilize all information (states, goals, and primitive skills) for composition, and our framework does leverage it and therefore, leads to better performance.
> > >
> > > AC: We would also like to thank our reviewer for providing some interesting ideas that we leave to our future studies such as i) leveraging variable input length at each time step, and ii) using hand-defined primitives.

---

> > > > ### Comment · AnonReviewer1 · 2019-11-15
> > > > **Paper has been clarified**
> > > >
> > > > Thank you for taking the time to clarify your work and adding additional experiments. I understand the method better and I think the paper has improved through this discussion. I am interested that this BiRNN-Attention architecture improves performance of combining skills, and agree that this is a valuable contribution. I will increase my score accordingly.

---

### Official Review · AnonReviewer3 · 2019-10-25
**Official Blind Review #3**

**Rating:** 6

**Review:**

This paper presents an approach in which new tasks can be solved by an attention model that can weigh the contribution of different base policies conditioned on the current state of the environment and task-specific goals. The authors demonstrate their method on a selection of RL tasks, such as an ant maze navigation task and a more complicated “ant fall” task, which requires the agent to first move a block to fill a gap in the environment before it is able to reach the goal.

I found the paper interesting and well written. My primary concern is that the primitive policies are learned independently of the composite policies, which might limit the application of this approach to more complex problems. Additionally, it would be great to also see the concurrent and sequential form of skill combination for the more complex tasks, and not just the point navigation task shown in Figure 7.

Standard errors on Figures 5 and 6 seem to be missing. Additionally, I was curious that in Figure 4a and Figure 6a, the composite’s performance is already a lot better than the other methods after 0 training steps. Maybe the authors can elaborate on that. Maybe the performance at step 0 is just hard to make out in the graphs?

I would suggest accepting the paper but there could be a more detailed analysis of how the pre-trained sub-modules are used and learning both composite and sub-policies together would make the paper stronger.

Additional comments:
- Where is the training graph for the Composite-SAV applied to the “ant fall” task? Maybe I missed it?
- Algorithm 1 should probably be moved to the main text.

####After rebuttal####
The authors' response and the revised paper address my already minor concerns.

**Experience Assessment:**

I have published one or two papers in this area.

**Review Assessment: Checking Correctness Of Derivations And Theory:**

I assessed the sensibility of the derivations and theory.

**Review Assessment: Checking Correctness Of Experiments:**

I assessed the sensibility of the experiments.

**Review Assessment: Thoroughness In Paper Reading:**

I read the paper thoroughly.

---

> ### Author Response · Authors · 2019-11-11
> **Response to Reviewer # 3**
>
> RC: reviewer comment; AR: author response
>
> RC: I found the paper interesting and well written. My primary concern is that the primitive policies are learned independently of the composite policies, which might limit the application of this approach to more complex problems.
>
> AR: We think it is one of the salient features of our method that it can take task-agnostic skills and can compose them for solving new transfer-learning problems. However, to address reviewer concern, we also include new skill acquisition experiments (Appendix B) in our paper. These experiments show that our method can acquire missing skills (task-dependent) and can compose them together with the existing task-agnostic skill set to solve the given problem in an end-to-end learning manner.
>
> RC: Additionally, it would be great to also see the concurrent and sequential form of skill combination for the more complex tasks, and not just the point navigation task shown in Figure 6.
>
> AR: We will add the depiction of attention weights for the other complicated tasks in a couple of days.
>
> RC: Standard errors on Figures 4 and 5 seem to be missing.
>
> AR: We have updated the figures to include standard errors. Please refer to the revised paper.
>
> RC: Additionally, I was curious that in Figure 4a and Figure 6a, the composite’s performance is already a lot better than the other methods after 0 training steps. Maybe the authors can elaborate on that. Maybe the performance at step 0 is just hard to make out in the graphs?
>
> AR: It is because other methods sometimes push the object away from the target position at early evaluation steps, leading to an increase in distance of the object from the given target than its initial distance.
>
> RC: I would suggest accepting the paper but there could be a more detailed analysis of how the pre-trained sub-modules are used and learning both composite and sub-policies together would make the paper stronger.
>
> AR: We will include the attention weights depiction soon for the other complicated tasks. Furthermore, we have added new results to show that it is possible to learn both composite and sub-level policies together in end-to-end learning using our framework.
>
> RC: Where is the training graph for the Composite-SAV applied to the “ant fall” task? Maybe I missed it?
>
> AR: We do not include as composite-SAC for ant-maze, ant-push, and ant-fall. These problems require complex task planning (sub-goal generation), and therefore, like other non-hierarchical RL methods (SAC, TRPO, PPO), the composition-SAC also fails to perform in these problems.
>
> RC: Algorithm 1 should probably be moved to the main text.
> AR: Due to limited space, it might not be possible to move the algorithm 1 to the main text.

---

> > ### Author Response · Authors · 2019-11-12
> > **Response to Reviewer# 3 [Attention Weights]**
> >
> > RC: Additionally, it would be great to also see the concurrent and sequential form of skill combination for the more complex tasks, and not just the point navigation task shown in Figure 6.
> >
> > AR:  We have included the depiction of attention weights for the other complicated environments in our paper (Appendix D).

---

### Author Response · Authors · 2019-11-15
**Response to all reviewers**

We thank all the reviewers for their insightful comments. We have revised our paper accordingly and provided the individual responses to each reviewer. The main changes to the paper are summarized as follows:

1-We add attention weights depiction for presented scenarios and update our plots to include std errors.
2-We revise our related work section to highlight our paper’s novelty and merits compared to concurrent work [1].
3-We include a couple of new experiments: i) a new ablated model (Sec 5.2) in our experiments (same as [1]) that takes the state and goal information, and outputs the mixture weights without conditioning on primitive actions encoding; ii) comparison with the benchmark models that were pretrained for an equal amount of time as our primitive policies (Fig. 4, Appendix C.1). iii) HIRO [2] results (Fig. 4) with both standard-mujoco-Ant (150 units torque limit) and low-torque-Ant (30 units torque limit, this is same as in [2]). iv) A side experiment (not the focus of this paper) that shows our method can learn new skills (that were missing in a given skill set) and composition simultaneously in an end-to-end manner (Appendix B).
All the experiments mentioned above further highlight the benefits of our approach. Our network architecture leverages all information (states, goals, primitive skills) in a structured way using bidirectional RNN and attention framework, and therefore, leads to better performance.

[1] “MCP: Learning Composable Hierarchical Control with Multiplicative Compositional Policies” NeurIPS 2019
[2] “Data-efficient hierarchical reinforcement learning” NeurIPS 2018

---

### Decision · Program_Chairs · 2019-12-19

**Decision:**

Accept (Poster)

**Comment:**

This paper considers deep reinforcement learning skill transfer and composition, through an attention model that weighs the contributions of several base policies conditioned on the task and state, and uses this to output an action. The method is evaluated on several Mujoco tasks.

There were two main areas of concern. The first was around issues with using equivalent primitives and training times for comparison methods. The second was around the general motivation of the paper, and also the motivation for using a BiRNN. These issues were resolved in a comprehensive discussion, leaving this as an interesting paper that should be accepted.